# Effects of vitamin $B_{12}$ supplementation on neurodevelopment and growth in Nepalese Infants: A randomized controlled trial

Tor A. Strand[1,2]*, Manjeswori Ulak[2,3], Mari Hysing[4], Suman Ranjitkar[3], Ingrid Kvestad[5], Merina Shrestha[3], Per M. Ueland[6], Adrian McCann[6], Prakash S. Shrestha[3], Laxman S. Shrestha[3], Ram K. Chandyo[7]

1 Innlandet Hospital Trust, Department of Research, Lillehammer, Norway, 2 Centre for International Health, University of Bergen, Bergen, Norway, 3 Department of Pediatrics, Child Health Research Project, Institute of Medicine, Tribhuvan University, Kathmandu, Nepal, 4 Department of Psychosocial Science, University of Bergen, Bergen, Norway, 5 Regional Center for Child and Youth Mental Health and Child Welfare, NORCE Norwegian Research Centre, Bergen, Norway, 6 Bevital AS, Bergen, Norway, 7 Department of Community Medicine, Kathmandu Medical College, Kathmandu, Nepal

* tor.strand@uib.no

## Abstract

### Background

Vitamin $B_{12}$ deficiency is common and affects cell division and differentiation, erythropoiesis, and the central nervous system. Several observational studies have demonstrated associations between biomarkers of vitamin $B_{12}$ status with growth, neurodevelopment, and anemia. The objective of this study was to measure the effects of daily supplementation of vitamin $B_{12}$ for 1 year on neurodevelopment, growth, and hemoglobin concentration in infants at risk of deficiency.

### Methods and findings

This is a community-based, individually randomized, double-blind placebo-controlled trial conducted in low- to middle-income neighborhoods in Bhaktapur, Nepal. We enrolled 600 marginally stunted, 6- to 11-month-old infants between April 2015 and February 2017. Children were randomized in a 1:1 ratio to 2 μg of vitamin $B_{12}$, corresponding to approximately 2 to 3 recommended daily allowances (RDAs) or a placebo daily for 12 months. Both groups were also given 15 other vitamins and minerals at around 1 RDA. The primary outcomes were neurodevelopment measured by the Bayley Scales of Infant and Toddler Development 3rd ed. (Bayley-III), attained growth, and hemoglobin concentration. Secondary outcomes included the metabolic response measured by plasma total homocysteine (tHcy) and methylmalonic acid (MMA). A total of 16 children (2.7%) in the vitamin $B_{12}$ group and 10 children (1.7%) in the placebo group were lost to follow-up. Of note, 94% of the scheduled daily doses of vitamin $B_{12}$ or placebo were reported to have been consumed (in part or completely). In this study, we observed that there were no effects of the intervention on the Bayley-III scores, growth, or hemoglobin concentration. Children in both groups grew on an average 12.5 cm (SD: 1.8), and the mean difference was 0.20 cm (95% confidence interval

**Data Availability Statement:** Data available on request. In order to meet ethical requirements for the use of confidential patient data, requests must

be approved by the Nepal Health Research Council (NHRC) and the Regional Committee for Medical and Health Research Ethics in Norway. Requests for data should be sent to the authors, by contacting NHRC (http://nhrc.gov.np), or by contacting the Department of Global Health and Primary Care at the University of Bergen (post@igs.uib.no).

**Funding:** This study was funded by grants from the Thrasher Research Fund (award # 11512), and the South-Eastern Norway Regional Health Authority (grant # 2012090). TAS reports funding from the South-Eastern Norway Regional Health Authority (grant # 2012090), IK from the Research Council of Norway (grant # 234495), and MU from Thrasher Research Fund (award # 11512) for conducting this research. AMC and PMU are paid employees at Bevital AS. The funders had no role in study design, data collection and analysis, decision to publish, or preparation of the manuscript.

**Competing interests:** I have read the journal's policy and the authors of this manuscript have the following competing interests: AMC and PMU are paid employees at Bevital AS.

**Abbreviations:** Bayley-III, Bayley Scales of Infant and Toddler Development 3rd ed.; CI, confidence interval; CONSORT, CONsolidated Standards of Reporting Trials; GC-MS/MS, gas chromatography-tandem mass spectrometry; GMR, geometric mean ratio; GSDF, geometric standard deviation factor; ICC, intra-class correlation coefficient; id, identification; IMCI, Integrated Management of Childhood Illness; MMA, methylmalonic acid; NHRC, Nepal Health Research Council; RDA, recommended daily allowance; RCT, randomized controlled trial; REC, Regional Committee for Medical and Health Research Ethics; tHcy, total Homocysteine; 3cB12, combined indicator of cobalamin status.

(CI): −0.23 to 0.63, $P = 0.354$). Furthermore, at the end of the study, the mean difference in hemoglobin concentration was 0.02 g/dL (95% CI: −1.33 to 1.37, $P = 0.978$), and the difference in the cognitive scaled scores was 0.16 (95% CI: −0.54 to 0.87, $P = 0.648$). The tHcy and MMA concentrations were 23% (95% CI: 17 to 30, $P < 0.001$) and 30% (95% CI: 15 to 46, $P < 0.001$) higher in the placebo group than in the vitamin $B_{12}$ group, respectively. We observed 43 adverse events in 36 children, and these events were not associated with the intervention. In addition, 20 in the vitamin $B_{12}$ group and 16 in the placebo group were hospitalized during the supplementation period. Important limitations of the study are that the strict inclusion criteria could limit the external validity and that the period of vitamin $B_{12}$ supplementation might not have covered a critical window for infant growth or brain development.

## Conclusions

In this study, we observed that vitamin $B_{12}$ supplementation in young children at risk of vitamin $B_{12}$ deficiency resulted in an improved metabolic response but did not affect neurodevelopment, growth, or hemoglobin concentration. Our results do not support widespread vitamin $B_{12}$ supplementation in marginalized infants from low-income countries.

## Trial registration

ClinicalTrials.gov NCT02272842

   **Universal Trial Number:** U1111-1161-5187 (September 8, 2014)

   **Trial Protocol:** Original trial protocol: PMID: 28431557 (reference [18]; study protocols and plan of analysis included as Supporting information).

## Author summary

### Why was this study done?

- Many marginalized children fail to reach their cognitive and growth potential.

- Subclinical vitamin $B_{12}$ deficiency, which is poor vitamin $B_{12}$ status without overt clinical symptoms, is common in this population in Nepal.

- Vitamin $B_{12}$ deficiency in children is associated with anemia (low hemoglobin concentration), stunted growth, and poor neurodevelopment.

### What did the researchers do and find?

- In this population-based, double-blind, randomized controlled trial (RCT), we measured the effects of daily supplementation of vitamin $B_{12}$ for 1 year in 600 infants.

- The primary outcomes were neurodevelopment, growth, and hemoglobin concentration.

- We targeted stunted infants as these children are at risk of vitamin $B_{12}$ deficiency.

- Daily supplementation of vitamin $B_{12}$ for a year resulted in a metabolic profile reflecting substantially improved $B_{12}$ status (lower total homocysteine (tHcy) and methylmalonic acid

(MMA) concentrations) but did not affect neurodevelopment, growth, or hemoglobin concentration.

## What do these findings mean?

- In spite of the improved metabolic profile following vitamin $B_{12}$ supplementation, the findings do not support widespread vitamin $B_{12}$ supplementation to improve short-term growth, neurodevelopment, or hemoglobin concentration in infants.

## Introduction

Vitamin $B_{12}$ (cobalamin) deficiency is common and affects all ages worldwide [1–3]. Animal-based foods are the primary sources, and poor status is prevalent in South Asia as well as in other low- and middle-income regions with low consumption of meat, animal milk, and fish [3,4].

Vitamin $B_{12}$ is required for cell division and differentiation, utilization of energy, and other critical metabolic processes [5–7]. Failure to thrive, delayed development, and macrocytic anemia are typical manifestations in children with severe deficiency [8]. Several observational studies have demonstrated associations between biomarkers of vitamin $B_{12}$ status with growth, neurodevelopment, and anemia [1,9–14]. The results from these studies suggest that the negative consequences of poor $B_{12}$ nutrition are also seen in suboptimal $B_{12}$ status, and not only in those with clinical deficiency. These observations, however, may be due to factors such as unmeasured confounding or limitations in the biomarkers used to assess vitamin $B_{12}$ status.

Three randomized controlled trials (RCTs) have measured the effect of vitamin $B_{12}$ supplementation on neurodevelopment in children with suboptimal $B_{12}$ status. Of these, 2 facility-based RCTs in infants born at low birth weight or who had developmental delay showed that a high dose of vitamin $B_{12}$ (400-μg hydroxycobalamin intramuscularly) substantially improved motor development [15,16]. In the third study, a population-based RCT in young North Indian children, there was a small borderline significant beneficial effect of daily oral supplementation with 1.8 μg of vitamin $B_{12}$ for 6 months on neurodevelopment [17]. The evidence from these RCTs support the notion that poor vitamin $B_{12}$ status may be a relevant public health concern; however, the evidence is currently not strong enough to alter widespread feeding recommendations. We therefore designed the current population-based RCT to measure the effects of daily supplementation of vitamin $B_{12}$ for 1 year. In this study, we ensured that the baseline $B_{12}$ status and metabolic response to the supplementation was well characterized, and we targeted marginally stunted infants due to their associated risk of vitamin $B_{12}$ deficiency, delayed development, and stunted growth.

## Participants and methods

### Study design and participants

This study is reported according to the CONsolidated Standards of Reporting Trials (CONSORT) guidelines (S1 Checklist). The study was a community-based, double-blind placebo-controlled trial in Nepalese infants. We hypothesized that daily administration of 2 μg vitamin $B_{12}$ for 12 months would improve neurodevelopment, growth, and hemoglobin concentration. We conducted the study in Bhaktapur municipality and surrounding peri-urban communities

near the capital Kathmandu. For the last 20 years, we have shown that poor vitamin $B_{12}$ status is common in both women and children in this population [4,18,19]. Bhaktapur is among the most densely populated municipalities in Nepal. The majority of births (96%) occur at health centers, and 1 in every second family is living in joint households and has a separate kitchen [20]. The primary outcomes were neurodevelopment measured by the Bayley Scales of Infant and Toddler Development 3rd ed. (Bayley-III), attained growth (cm and kg), and hemoglobin concentration. The secondary outcome was the metabolic response measured by plasma concentrations of cobalamin, total homocysteine (tHcy), and methylmalonic acid (MMA).

## Ethics statement

The study received ethical clearance from the Nepal Health Research Council (NHRC; #233/2014) and from the Regional Committee for Medical and Health Research Ethics (REC; #2014/1528) in Norway. After detailed information was provided to parents, we obtained written informed consent or a thumbprint from those who were illiterate (in the presence of an impartial witness).

## Enrollment, randomization, and blinding

We enrolled 600 children between April 17, 2015 and February 15, 2017. Inclusion criteria were age 6 to 11 months, length for age z-score $<-1$, intent to reside in the municipality and surrounding areas for the next 12 months, and availability of informed consent from parents. Children were excluded if they were taking (or planned to take) supplements that contained vitamin $B_{12}$, had a severe systemic illness requiring hospitalization, if they were severely malnourished (weight for length z-score $<-3$), were severely anemic (hemoglobin concentration $<7$ g/dL), or had ongoing infections that required medical treatment. In cases of severe malnutrition, anemia, or infections, children received treatment and were screened again for eligibility after recovery.

Field workers identified eligible children from immunization clinics or through home visits. Infants were enrolled by a study supervisor or by a physician at the field office. We randomized the infants in a 1:1 ratio in blocks of 8 using a computer-generated randomization list. Randomization was concealed, and the study double-blinded as the participants were only linked to the intervention through the identification (id) number printed on the supplement labels. The list that linked this id number to the randomization code was kept with the producers of the supplements and the scientist who generated it. None of the investigators had access to this list until the data collection and cleaning for the primary outcomes were completed. The vitamin $B_{12}$ supplements and placebo were produced specifically for the trial and were identical in taste and appearance. At enrollment and end of study, we measured neurodevelopment, weight, length, and hemoglobin concentration.

## Intervention and co-interventions

All children received 2 μg of vitamin $B_{12}$ (cyanocobalamin), corresponding to approximately 2 to 3 recommended daily allowances (RDAs) or placebo via a daily oral supplement for 12 months. The intervention was implemented using sachets containing 20 grams of a lipid-based paste produced by GC Rieber Compact (Gurgaon, Haryana, India; http://www.gcrieber-compact.com/). Each sachet provided the daily dose of supplements. To ensure that the effect of vitamin $B_{12}$ was not limited by inadequate intake of other essential nutrients, both the placebo and vitamin $B_{12}$ paste contained a base multi-micronutrient mixture with several other vitamins and minerals at approximately 1 RDA. All caregivers were given dietary recommendations according to national guidelines. Children who developed diarrhea during the

intervention period received zinc and oral rehydration solution. Those with mild to moderate anemia (hemoglobin 7 to 10 g/dL) were treated with per oral iron for at least 30 days. Children with pneumonia, dysentery, or other illnesses were treated according to the most recent Integrated Management of Childhood Illness (IMCI) guidelines [21].

During weekly visits to the homes, field workers asked the mothers about intake of the paste during the past 7 days and recorded in detail the amount of paste given to the children (i.e., half, one-third, three-fourths, or less). All episodes of vomiting or regurgitation after supplementation of the paste were also recorded, and the total number of empty paste sachets were counted at the weekly visits to verify the reported compliance.

## Outcomes

### Neurodevelopment

The Bayley-III is a comprehensive assessment tool of neurodevelopment in infants and toddlers aged 1 to 42 months [22]. The Bayley-III is often regarded as the gold standard for assessing neurodevelopment in this age-group and is used in research worldwide. We administered the Bayley-III directly with the child at enrollment and end of the study at the study research office. The Bayley-III consists of a cognitive, language (receptive and expressive), motor (fine and gross motor), and socio-emotional scale. Three psychologists, of whom 1 had extensive experience with the Bayley-III, were trained to perform the assessments for the study. To ensure high-quality measurements, we performed standardization exercises in 20 children ahead of the enrollment where the Bayley assessments were scored by 2 raters. The psychologists were required to reach an intra-class correlation coefficient (ICC) >0.90 with the expert rater, who served as the gold standard. Of note, 7% of the sessions during the main study were double scored by the expert rater to ensure appropriate interobserver agreement throughout the study. The ICCs from the quality controls ranged from 0.95 to 0.99 [23]. All the Bayley-III assessments were video-recorded for quality purposes. The Bayley-III raw scores were converted into scaled and composite scores based on U.S. citizen normative data [22]. For the analyses, we used raw, scaled, and composite scores.

### Anthropometry

Weight was measured with a portable electronic scale (model 877, Seca, California, United States of America) that measures to the nearest 0.01 kg, and length was measured with portable length board (model 417, Seca, California, USA). We measured length and weight at the clinic or at home during the monthly follow-up visits. All anthropometric measurements were performed twice. The mean values were used in the analyses.

### Laboratory procedures

Blood samples were collected from the cubital veins into polypropylene tubes containing EDTA (Sarstedt, Germany), which were protected from direct sunlight exposure. Up to 4-mL blood was collected at enrollment and end of study. The hemoglobin concentration was analyzed immediately following blood sampling with HemoCue (HemoCue 201, Ångelholm, Sweden), which was calibrated as per the guidelines defined by the manufacturer. The blood was centrifuged at room temperature for 10 min at 2,000 to 2,500 g within 10 min after venipuncture (Model R-304, Remi, Mumbai, India). Plasma and blood cells were separated, transferred into polypropylene vials (Eppendorf, Germany), and immediately stored at <−80˚C until analysis at Bevital Laboratory (Bergen, Norway; www.bevital.no). The samples were shipped to Norway on dry ice by World Courier. The plasma concentrations of cobalamin and folate

were determined using microbiological assays [24,25] using a colistinsulfate-resistant strain of *Lactobacillus leichmannii* or chloramphenicol-resistant strain of *Lactobacillus casei*, respectively. The functional biomarkers plasma total tHcy and MMA are considered to be sensitive markers of $B_{12}$ deficiency [26]. Plasma tHcy and MMA were analyzed by gas chromatography-tandem mass spectrometry (GC-MS/MS) based on methylchloroformate derivatization [27]. The within-day coefficient of variation was 4% for both cobalamin and folate and ranged from 1% to 5% for tHcy and MMA. The between-day coefficient of variation was 5% for both cobalamin and folate and ranged from 1% to 8% for MMA and tHcy. We also calculated a combined indicator of cobalamin status (3cB12) based on the 3 biomarkers cobalamin, tHcy, and MMA as suggested by Fedosov and colleagues [28]. In short, this index is the log of the cobalamin concentration divided by the product of the log of the tHcy and MMA concentrations.

## Sample size

The study had 80% power to detect a standardized effect size of 0.22 and had 90% power to detect an effect size of 0.28. In these calculations, we assumed a loss to follow-up of 10%. Details on the samples size calculations are also presented in the previously published protocol paper [18].

## Statistical analyses

The analyses in this study were carried out according to the predefined protocols and analysis plans (S1–S4 Texts). Weight for age, weight for length, and length for age z-scores were calculated using the most recent WHO growth charts [29]. We defined underweight, stunting, and wasting as z-scores below −2 [29]. We depicted the relationship between changes in vitamin $B_{12}$ status (i.e., change in the 3cB12 values from baseline to end of study) according to vitamin $B_{12}$ status (3cB12) at baseline. To do so, we performed a kernel-weighted local polynomial regression of delta 3cB12 on baseline 3cB12 values by treatment group and depicted these dose-response graphs of the predicted values with 95% confidence intervals (CIs). We also compared the concentrations of cobalamin, tHcy, and MMA between the study groups in all infants and according to 3cB12 categories ("possible deficient," "low," and "adequate") at baseline. As these variables were left-skewed, we present the geometric means and the geometric standard deviation factors (GSDFs). We used the log-transformed values of the end-study biomarker concentrations when comparing the differences between the study groups. The exponentials of these mean differences are presented as the geometric mean ratio (GMR) between the study groups.

We compared the mean end-study Bayley-III scaled and composite scores between the intervention groups. For the other outcome variables (Bayley-III raw scores, growth, and hemoglobin concentrations), the effects of the intervention were estimated by comparing the changes from baseline to end study between the study groups. The precision of the effect estimates and corresponding *P* values were calculated using the Student *t* test assuming equal variances. We also analyzed the data by several predefined subgroups. The subgrouping variables were stunting (defined as length for age z-scores $<-2$), underweight (weight for age z-scores $<-2$), low vitamin $B_{12}$ status (3cB12 $<-0.5$), low birth weight (birth weight $<2,500$ g), anemia (hemoglobin concentration $<11.0$ g/dL), and exclusive breastfeeding. For these analyses, we used univariate and multiple generalized linear models with the Gaussian distribution family and identity link function adjusting for a set of predefined potential confounders. The following variables were adjusted for in all subgroup analyses: length for age z-scores, maternal and paternal education, and age of the child at baseline. All analyses were performed using Stata version 16 (StataCorp, College Station, Texas, USA).

## Results

From April 2015 to February 2017, we screened 733 infants and randomized 600 into the study (Fig 1). A total of 26 infants dropped out due to refusal or migration. Two children were unable to complete the end-study activities, leaving 572 infants with complete neurodevelopmental assessments. We were able to collect and analyze blood samples for biomarker assessments at end study from 567 children.

Baseline features by intervention and placebo groups are presented in Table 1. One in every five infants was born at low birth weight, and one-third were stunted ($<-2$ z-score length for age) at enrollment. The baseline features were evenly distributed between the intervention groups. The baseline status of vitamin $B_{12}$ and its plasma biomarkers were also comparable between the intervention groups (S1 Table).

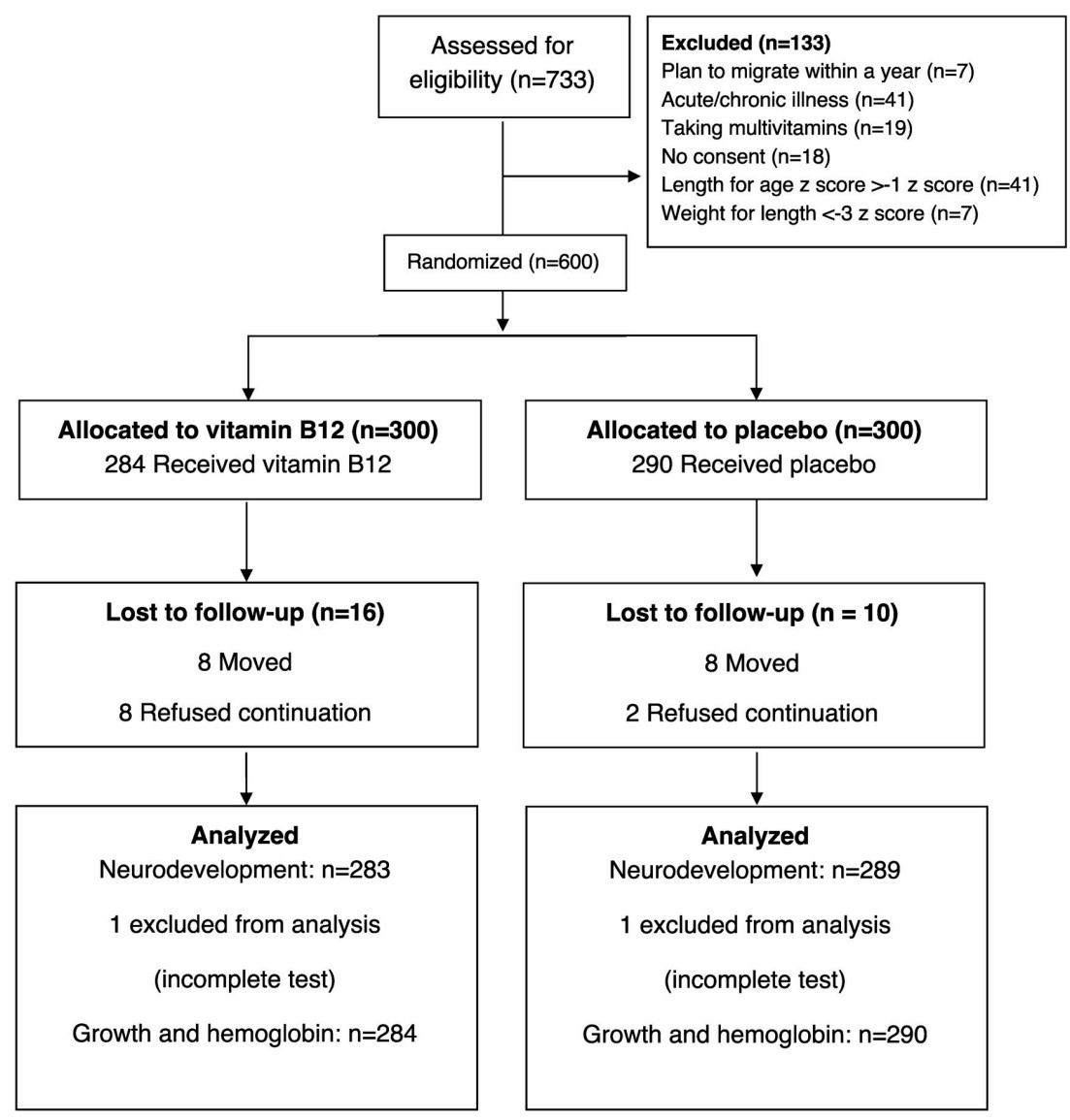

**Fig 1. Trial flowchart of a study measuring the effect of daily vitamin $B_{12}$ supplementation in Nepalese infants.**

**Table 1. Baseline characteristics in a study investigating the effect of daily vitamin $B_{12}$ supplementation on neurodevelopment and growth in 600 Nepalese infants.**

| | Vitamin $B_{12}$ group ($n$ = 300) | | Placebo group ($n$ = 300) | |
|---|---|---|---|---|
| | $n$ | % | $n$ | % |
| **Infant characteristics** | | | | |
| Mean age of child (months), mean ± SD | 8.1 ± 1.7 | | 8.0 ± 1.8 | |
| Male child | 158 | 53 | 151 | 50 |
| Have older siblings | 155 | 52 | 153 | 51 |
| Low birth weight (<2,500 gm)[1] | 56 | 19 | 59 | 20 |
| Hospitalization within first month of age | 28 | 9.3 | 26 | 8.7 |
| **Demographic features** | | | | |
| Mother's age, mean ± SD | 27.1 ± 4.7 | | 27.5 ± 4.6 | |
| Father's age[2], mean ± SD | 30.0 ± 7.1 | | 30.6 ± 5.1 | |
| Mothers who completed secondary school or above | 197 | 65.7 | 180 | 60 |
| Fathers who completed secondary school or above | 199 | 66.3 | 189 | 63 |
| Mothers who work | 117 | 39.0 | 110 | 36.7 |
| Fathers who work | 286 | 95.3 | 280 | 93.3 |
| **Socioeconomic status** | | | | |
| Family staying in joint family | 143 | 47.7 | 149 | 49.7 |
| Family residing in rented house | 152 | 50.7 | 139 | 46.3 |
| Number of rooms in use by the household (<−2) | 163 | 54.3 | 174 | 58 |
| Kitchen and bedroom in the same room | 148 | 49.3 | 150 | 50 |
| Family having own land | 138 | 46 | 144 | 48 |
| Receiving remittance from abroad | 30 | 10 | 27 | 9 |
| **Breastfeeding status** | | | | |
| No breastfeeding at time of interview | 8 | 2.7 | 6 | 2 |
| Exclusive breastfeeding for 3 months or more | 143 | 47.7 | 137 | 45.6 |
| **Nutritional status of infants** | | | | |
| Underweight (weight for age z-score <−2) | 62 | 20.7 | 50 | 16.6 |
| Stunting (length for age z-score <−2) | 96 | 32.1 | 98 | 32.7 |
| Wasting (weight for length z-score <−2) | 12 | 4.0 | 7 | 2.3 |
| Hemoglobin, g/dL, mean ± SD | 10.6 ± 0.96 | | 10.6 ± 0.91 | |
| Anemia (hemoglobin <11 g/dL) | 183 | 61 | 202 | 67.3 |
| **Nutritional status of mother** | | | | |
| BMI of mother, mean ± SD | 23.7 ± 3.5 | | 23.7 ± 3.6 | |
| <18.5 kg/m² BMI of mother | 15 | 5 | 19 | 6.3 |

[1]Among 579 infants whose birth weights were recorded.

[2]Among 487 fathers who were available.

$n$, number.

## Compliance

More than 94% of the prescribed doses were reportedly taken (94.3% in the vitamin $B_{12}$ group and 94.0% in the placebo group; S2 Table). Of these, 86.3% and 84.7% of the vitamin $B_{12}$ and placebo group, respectively, consumed the entire prescribed doses.

## End points

Vitamin $B_{12}$ supplementation had no effect on any of the neurodevelopmental outcomes (Table 2). For example, the cognitive composite scores were 0.73 points (95% CI: −0.55 to 2.02,

**Table 2. Effect of daily vitamin B$_{12}$ supplementation for 1 year starting in infancy on the Bayley Scales of Infant Development scores among infants in Bhaktapur, Nepal.**

| Bayley-III subscales | Vitamin B$_{12}$ group (n = 283) | Placebo group (n = 289) | Mean differences |
|---|---|---|---|
| | Mean ± SD | Mean ± SD | (95% CI) (P value) |
| **Change in raw scores from baseline until end study** | | | |
| Cognitive | 20.9 ± 4.3 | 20.7 ± 4.2 | 0.16 (−0.54 to 0.87) (0.648) |
| Language | | | |
| Expressive | 14.4 ± 4.7 | 14.8 ± 5.0 | 0.35 (−0.44 to 1.14) (0.387) |
| Receptive | 10.9 ± 4.0 | 11.2 ± 3.6 | −0.31 (−0.94 to 0.32) (0.334) |
| Motor | | | |
| Fine motor | 13.8 ± 3.3 | 13.7 ± 2.9 | 0.10 (−0.41 to 0.60) (0.712) |
| Gross motor | 21.4 ± 5.1 | 21.7 ± 4.7 | −0.28 (−1.08 to 0.52) (0.491) |
| Socio-emotional | 29.4 ± 14.3 | 31.0 ± 12.0 | −1.57 (−3.96 to 0.82) (0.197) |
| **End-study composite and scaled score** | | | |
| Cognitive composite score | 90.5 ± 8.2 | 91.2 ± 7.3 | 0.73 (−0.55 to 2.02) (0.261) |
| Language composite score | 93.0 ± 12.8 | 92.6 ± 12.6 | −0.42 (−2.51 to 1.66) (0.692) |
| Expressive scaled score | 8.6 ± 2.6 | 8.5 ± 2.4 | −0.01 (−0.43 to 0.41) (0.965) |
| Receptive scaled score | 9.0 ± 2.3 | 8.9 ± 2.5 | −0.13 (−0.53 to 0.27) (0.517) |
| Motor composite score | 99.9 ± 8.7 | 100.2 ± 8.3 | 0.32 (−1.08 to 1.73) (0.652) |
| Fine motor scaled score | 10.7 ± 1.6 | 10.9 ± 1.8 | 0.22 (−0.06 to 0.50) (0.131) |
| Gross motor scaled score | 9.2 ± 2.0 | 9.1 ± 1.7 | −0.12 (−0.42 to 0.19) (0.457) |
| Socio-emotional composite score | 104.3 ± 16.7 | 103.4 ± 17.1 | −0.92 (−3.70 to 1.86) (0.518) |

The mean differences, the corresponding 95% CI, and P values were calculated using Student t test assuming equal variances.

Bayley-III, Bayley Scales of Infant and Toddler Development 3rd ed.; CI, 95% confidence interval; n, number.

P = 0.261) lower in the vitamin B$_{12}$ group compared to the placebo group. There was also no effect on growth or hemoglobin concentration (Table 3). Children in both groups grew on an average 12.5 cm (SD: 1.8), and the mean difference was 0.20 cm (95% CI: −0.23 to 0.63, P = 0.354). The mean difference in hemoglobin concentration between the groups was 0.02 g/dL (95% CI: −1.33 to 1.37, P = 0.978). The adjusted effects of the interventions on growth and neurodevelopment by

**Table 3. Effect of vitamin B$_{12}$ supplementation on growth and hemoglobin concentrations among infants in Bhaktapur, Nepal.**

| | Vitamin B$_{12}$ group (n = 283) | Placebo group (n = 290) | Mean differences |
|---|---|---|---|
| | Mean ± SD | Mean ± SD | (95% CI) (P value) |
| **Change from baseline** | | | |
| Length (cm) | 12.5 ± 1.8 | 12.5 ± 1.8 | 0.09 (−0.21 to 0.39) (0.574) |
| Weight (kg) | 2.1 ± 0.5 | 2.1 ± 0.6 | −0.01 (−0.10 to 0.09) (0.919) |
| Hemoglobin (g/dL) | 1.0 ± 1.1 | 1.0 ± 1.2 | −0.02 (−0.20 to 0.17) (0.876) |
| **End study** | | | |
| Length (cm) | 78.2 ± 2.6 | 78.4 ± 2.6 | 0.20 (−0.23 to 0.63) (0.354) |
| Weight (kg) | 9.4 ± 0.9 | 9.4 ± 1.0 | −0.02 (−0.18 to 0.13) (0.780) |
| Length for age z-score | −1.8 ± 0.7 | −1.7 ± 0.7 | 0.05 (−0.06 to 0.16) (0.411) |
| Weight for height z-score | −0.7 ± 0.8 | −0.8 ± 0.8 | −0.08 (−0.22 to 0.05) (0.238) |
| Weight for length z-score | −1.4 ± 0.7 | −1.4 ± 0.8 | −0.04 (−0.16 to 0.08) (0.529) |
| Hemoglobin (g/dL) | 11.6 ± 0.8 | 11.6 ± 1.0 | 0.02 (−1.33 to 1.37) (0.978) |

The mean differences, the corresponding 95% CI, and P values were calculated using Student t test assuming equal variances.

CI, 95% confidence interval

various subgroups are shown in S1 and S2 Figs. These analyses did not reveal any variable that modified the effect on any of the outcomes. Unadjusted subgroup analyses yielded the same results. We found no adverse effects of vitamin $B_{12}$ supplementation (S3 Table).

The effect of vitamin $B_{12}$ supplementation on $B_{12}$ status expressed by the 3cB12 according to baseline status is depicted in Fig 2. The distance between the solid and dotted lines represents the metabolic effect of the intervention, which decreased as the baseline vitamin $B_{12}$ status improved. A similar trend is displayed in S1 Table where the effects on the different biomarkers are shown. For the functional biomarkers, tHcy and MMA, poorer vitamin $B_{12}$ status at baseline was associated with a larger effect of vitamin $B_{12}$ supplementation. For example, when restricting the analyses to infants who were classified as possibly deficient, the placebo group had 60% higher MMA concentrations (indicating poorer vitamin $B_{12}$ status) at end study compared to those in the vitamin $B_{12}$ group (GMR 1.60, 95% CI: 1.20 to 2.14, $P < 0.001$). In the children who had adequate status at baseline, there was no effect of vitamin $B_{12}$ supplementation on the MMA concentration (S1 Table).

## Discussion

In this year-long, double-blind, placebo-controlled RCT, daily intake of a supplement containing vitamin $B_{12}$ improved vitamin $B_{12}$ status in marginally stunted Nepalese infants. Those with poor status at the onset of the study benefitted more than those with adequate status. The intervention, however, did not result in any improvements in neurodevelopment, nor was there any effects on growth or hemoglobin concentration. Restricting the analyses to those with poor status at baseline, i.e., those who also had the best metabolic response, did not alter these results.

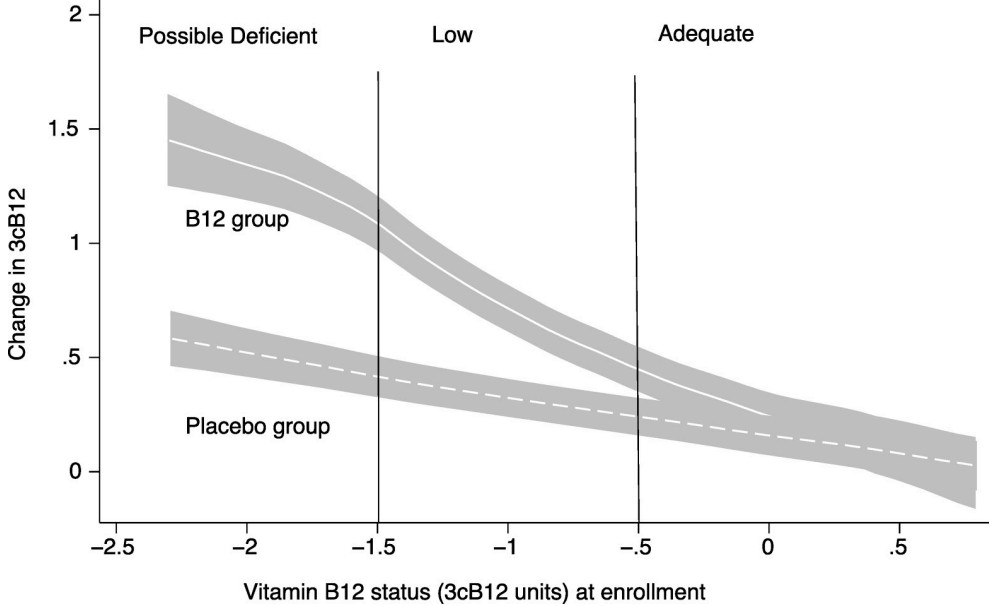

**Fig 2. The association between vitamin $B_{12}$ status at baseline and change in vitamin $B_{12}$ status from baseline to end study by randomized group.** The y-axis is the change in 3cB12 from baseline to end study, and the x-axis is the baseline 3cB12 value. The regression lines were generated by a kernel-weighted local polynomial regression with the change in 3cB12 as the dependent variable and 3cB12 at baseline as the independent variable. The shaded areas represent the 95% CI of the regression lines. The profiles were generated separately for the 2 intervention groups, and the distance between the 2 regression lines represents the effect of the vitamin $B_{12}$ supplementation. The 3cB12 is a function of the plasma concentrations of cobalamin, tHcy, and MMA. CI, 95% confidence interval; MMA, methylmalonic acid; tHcy, total Homocysteine.

Our findings are in contrast to results from the observational studies where vitamin $B_{12}$ status was positively associated with neurodevelopment, growth, and hemoglobin concentration [10,14,30]. The findings are also at odds with results from the 3 small (group sizes of 32 to 104) RCTs in infants and young children where vitamin $B_{12}$ supplementation resulted in improved developmental scores [15–17].

Several elements of our study design and conduct support the veracity of our findings. The trial enrolled 600 children, of whom >95% could be included in the analyses. As a result, we have precise effect estimates of our primary outcomes, and thus, sufficient power to detect clinically meaningful differences between the treatment groups. The randomization was successful as indicated by no differences in baseline characteristics, and few participants were lost to follow-up. Finally, the compliance with supplement use was high, as it was given on nearly 95% of the scheduled days and resulted in an excellent metabolic response (S2 Table).

The study team has extensive experience with the Bayley-III, and the inter-rater reliability was excellent during both the initial standardization exercises as well as for the quality controls throughout the study. The Bayley-III scores were associated with established risk factors for poor neurodevelopment [23], which provided support for the validity of the test in this study setting. In addition, the study staff were trained and standardized for 2 decades in measuring infant growth, ensuring precise measurements for these outcomes. Lastly, we had an effective cold chain and used state-of-the-art biochemical methods to estimate the biomarker concentrations ensuring optimal description of the vitamin status.

The participants were moderately stunted infants and accordingly at risk of poor neurodevelopment and vitamin $B_{12}$ deficiency. Thus, they constituted a group of children where we could expect an effect of vitamin $B_{12}$ supplementation if subclinical deficiency affected any of our outcomes. For practical and ethical reasons, we could not target children with defined or overt $B_{12}$ deficiency. We did not have the necessary diagnostic recourses to measure vitamin $B_{12}$ status before randomization, and giving a placebo to infants with diagnosed vitamin $B_{12}$ deficiency for a year would violate the principle of clinical equipoise. Thus, the study included some children with adequate vitamin $B_{12}$ status at baseline. It is plausible that inclusion of vitamin $B_{12}$ replete children could attenuate a potential effect of the intervention. However, this does not explain our null findings as the effect estimates did not change when we restricted the analyses to those with evidence of deficiency.

A higher dose or a different, more effective, mode of administration (i.e., injection) could have led to different results. The metabolic response, however, indicates a substantial biological effect. Vitamin $B_{12}$ is involved in 2 biochemical reactions in humans [31]. In 1 of these, vitamin $B_{12}$ is required for the transfer of methyl groups, which includes remethylation of homocysteine to methionine. Disruption of this pathway increases homocysteine and affects gene regulation and DNA synthesis [31]. Vitamin $B_{12}$ also acts as an enzymatic cofactor for methylmalonyl-CoA mutase, an enzyme involved in the catabolism of fats and amino acids. Disruption of the latter pathway explains the increased MMA observed in vitamin $B_{12}$ deficiency. Our study provides causal evidence that both of these metabolic pathways are affected by mild vitamin $B_{12}$ deficiency in children, and their functioning improves with supplementation. Thus, despite no effects on the clinical outcomes, children in this population could benefit from increasing the vitamin $B_{12}$ intake as indicated by their improved metabolic profile.

The period of supplementation in the present study might not have covered a critical window where adequate vitamin $B_{12}$ status is crucial for optimal growth and neurodevelopment. Thus, initiating supplementation sooner, such as before or during pregnancy or earlier in infancy, could have yielded different results.

Restricting the participants to mildly stunted children increased the internal validity and statistical power of our study at the expense of the external validity. In other words, our

recruitment strategy reduced the generalizability of our results, which is a limitation. It should be noted that enrolling all infants from the community would result in a larger proportion with adequate vitamin $B_{12}$ status and, consequently, more children who would not respond to the intervention. A design ensuring higher external validity could accordingly attenuate a potential effect and, at the same time, increase the variability, both contributing to reduced statistical power. Reduced statistical power increases the risk of false-negative results (type II errors), which particularly question null findings such as here. In addition, our study was designed to measure the effects of vitamin $B_{12}$ supplementation on several outcomes across several subgroups, which reduced the probability of overlooking relevant short-term clinical outcomes.

An important assumption of our study was that the inclusion criteria ensured we targeted those who would benefit from supplementation. It is possible, however, that these stunted children would suffer from deficiencies of other growth-limiting nutrients. We believe this potential bias was accounted for by providing other vitamins and minerals, by treating common infections, and by giving zinc for diarrhea.

In summary, our results show that in Nepalese infants, daily vitamin $B_{12}$ supplementation improves vitamin $B_{12}$ status and the metabolic profile expressed by the composite 3cB12 indicator. The metabolic response indicates that this population could benefit from vitamin $B_{12}$ supplementation, but the clinical consequences of subclinical deficiency in early childhood remain uncertain.

## Supporting information

**S1 Fig. The effect of vitamin $B_{12}$ supplementation on the Bayley Scales of Infant Development subscale scores in different subgroups.** A point estimate to the left of the vertical line indicates a beneficial effect of vitamin $B_{12}$. None of the subgroup specific effects were statistically significant. The effect estimates were calculated with multiple general linear models with the Gaussian distribution family and identity link function adjusting for length for age z-scores, maternal and paternal education, and age of the child at baseline. Stunting and underweight were defined as being $<-2$ length for age z-scores and weight for age z-scores, respectively. 3cB12: combined vitamin $B_{12}$ status indicator as suggested by Fedosov and colleagues [28], low 3cB12 is $<-0.5$, low birth weight: birth weight $<2,500$ g, anemia: hemoglobin concentration $<11$ g/dL.
(TIF)

**S2 Fig. The effect of vitamin $B_{12}$ supplementation on growth and hemoglobin concentration in different subgroups.** A point estimate to the left of the vertical line indicates a beneficial effect of vitamin $B_{12}$. None of the subgroup specific estimates were statistically significant. The effect estimates were calculated with multiple general linear models with the Gaussian distribution family and identity link function adjusting for length for age z-scores, maternal and paternal education, and age of the child at baseline. Stunting and underweight were defined as being $<-2$ length for age z-scores and weight for age z-scores, respectively. 3cB12: combined vitamin $B_{12}$ status indicator as suggested by Fedosov and colleagues [28], low 3cB12 is $<-0.5$, low birth weight: birth weight $<2,500$ g, anemia: hemoglobin concentration $<11$ g/dL.
(TIF)

**S1 Table. Effects of daily vitamin $B_{12}$ supplementation for 1 year starting in infancy on markers of vitamin $B_{12}$ status.**
(DOCX)

**S2 Table. Compliance of vitamin B$_{12}$ supplementation among Nepalese infants participating in clinical trial on the effect of vitamin B$_{12}$ supplementation on growth, development, and hemoglobin concentration.**
(DOCX)

**S3 Table. Adverse effects of vitamin B$_{12}$ supplementation among Nepalese infants participating in clinical trial on the effect of vitamin B$_{12}$ supplementation on growth, development, and hemoglobin concentration.**
(DOCX)

**S1 Text. Main protocol version 1.0.** August 1, 2014.
(DOCX)

**S2 Text. Main protocol version 2.1.** October 16, 2016.
(DOCX)

**S3 Text. Plan of analysis version 1.** March 2018.
(DOCX)

**S4 Text. Plan of analysis version 2.1.** October 2019.
(DOCX)

**S1 Checklist. CONSORT Checklist.**
(DOCX)

## Acknowledgments

We would like to express our gratitude to all the field staff, children, and families in Bhaktapur who participated in the study. We are also grateful to the Child Health Research Project Team at the Department of Child Health at the Institute of Medicine, Tribhuvan University and Siddhi Memorial Foundation and its founder Shyam Dhaubhadel. Finally, we thank Johanne Haugen who was responsible for randomization.

## Author Contributions

**Conceptualization:** Tor A. Strand, Manjeswori Ulak, Mari Hysing, Ingrid Kvestad, Prakash S. Shrestha, Ram K. Chandyo.

**Formal analysis:** Tor A. Strand, Manjeswori Ulak, Mari Hysing, Suman Ranjitkar, Ingrid Kvestad, Adrian McCann, Ram K. Chandyo.

**Funding acquisition:** Tor A. Strand, Mari Hysing, Ingrid Kvestad, Prakash S. Shrestha, Ram K. Chandyo.

**Investigation:** Manjeswori Ulak, Mari Hysing, Suman Ranjitkar, Ingrid Kvestad, Merina Shrestha, Per M. Ueland, Adrian McCann, Laxman S. Shrestha, Ram K. Chandyo.

**Methodology:** Manjeswori Ulak, Mari Hysing, Ingrid Kvestad, Per M. Ueland, Ram K. Chandyo.

**Project administration:** Tor A. Strand, Manjeswori Ulak, Laxman S. Shrestha, Ram K. Chandyo.

**Supervision:** Tor A. Strand, Mari Hysing, Suman Ranjitkar, Ingrid Kvestad, Merina Shrestha, Laxman S. Shrestha, Ram K. Chandyo.

**Validation:** Mari Hysing, Ingrid Kvestad, Ram K. Chandyo.

**Writing – original draft:** Tor A. Strand, Manjeswori Ulak, Mari Hysing, Suman Ranjitkar, Ingrid Kvestad, Merina Shrestha, Per M. Ueland, Adrian McCann, Laxman S. Shrestha, Ram K. Chandyo.

**Writing – review & editing:** Manjeswori Ulak, Mari Hysing, Suman Ranjitkar, Ingrid Kvestad, Merina Shrestha, Per M. Ueland, Adrian McCann, Prakash S. Shrestha, Laxman S. Shrestha, Ram K. Chandyo.

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
