## [Editor Report · Decision Letter 0]

11 Feb 2020

Dear Dr Strand, 

Thank you for submitting your manuscript entitled "A randomized controlled trial of vitamin B12 supplementation, neurodevelopment, and growth in Nepalese infants" for consideration by PLOS Medicine.

Your manuscript has now been evaluated by the PLOS Medicine editorial staff, as well as by an academic editor with relevant expertise, and I am writing to let you know that we would like to send your submission out for external peer review.

Kind regards,

Helen Howard, for Clare Stone PhD 

Acting Editor-in-Chief

PLOS Medicine 

plosmedicine.org

---

## [Decision Letter · Decision Letter 1]

27 Aug 2020

Dear Dr. Strand,

Thank you very much for submitting your manuscript "A randomized controlled trial of vitamin B12 supplementation, neurodevelopment, and growth in Nepalese infants" (PMEDICINE-D-20-00074R1) for consideration at PLOS Medicine. 

[LINK]

In light of these reviews, I am afraid that we will not be able to accept the manuscript for publication in the journal in its current form, but we would like to consider a revised version that addresses the reviewers' and editors' comments. Obviously we cannot make any decision about publication until we have seen the revised manuscript and your response, and we plan to seek re-review by one or more of the reviewers. 

We expect to receive your revised manuscript by Sep 17 2020 11:59PM. Please email us (plosmedicine@plos.org) if you have any questions or concerns.

We look forward to receiving your revised manuscript. 

Sincerely,

Emma Veitch, PhD

PLOS Medicine

On behalf of Clare Stone, PhD, Acting Chief Editor, 

PLOS Medicine

plosmedicine.org

*If possible, please present the title per the PLOS Medicine style, this should present the study objective/question in the first part of the title and then the study design (eg, "A randomized controlled trial," "A retrospective study," "A modelling study," etc.) in the subtitle (ie, after a colon).

*In the last sentence of the Abstract Methods and Findings section, please include a brief note about any key limitation(s) of the study's methodology.

*Please complete the CONSORT checklist (http://www.consort-statement.org/), designed to aid reporting of randomized trials, and ensure that all components of CONSORT are present in the manuscript. We'd suggest noting in the methods section that the trial is reported in line with CONSORT; please upload the completed CONSORT checklist as a supplementary file with your resubmission.

*Per the CONSORT guidance, we'd recommend that the abstract and Results section should present the estimates of effect size (with 95% CI's) on the main outcomes for the trial so that readers can get a sense of what the range of effects are that are plausible with the data - rather than just the narrative summary that there is no effect. 

*It would be helpful if the authors could say something about possible harms (adverse effects) with the supplementation in the trial - the trial registration suggests that adverse effects would be monitored but the results section does not seem to present any data for this. Obviously in trials both benefits and risks are important.

*It would also be good to include some discussion about possible external validity (generalisability) of the trial findings given that, as noted by reviewers, the children in the study are marginally stunted. 

Comments from the reviewers:

Reviewer #1: This is a statistical review of manuscript PMEDICINE-D-20-00074_R1. The manuscript is well written and easy to follow. Some clarifications are needed in the statistical section. 

Major comments: 

* Statistical analyses section: "Differences in proportions were calculated using generalized linear models (GLM) with the binomial distribution family and identity link function". Please could you clarify with you did not use one of the most common link functions for binary data (i.e. logit, cloglog, etc). The identity link is rarely used. 

* Statistical analyses section: "For these analyses we used crude and multiple GLMs with the gaussian distribution family and identity link function adjusting for a set of predefined potential confounders". Are you referring to continuous endpoints or binary endpoints? It is not clear currently, as this sentence follows the previous sentence that cover difference in proportions. 

Minor comment:

* Abstract: please specify what was percentage of loss to follow up (26/600=4.3%). Also, 2 children could not be included in the analysis. I think it might be worth stating the size of your complete data set and stating how it compares to the original 600 children as percentage. 

Reviewer #2: This is a clearly written paper describing a blinded RCT of Vitamin B12 supplementation in Nepalese infants. The primary outcomes were neurodevelopment, growth and haemoglobin concentration. The authors have succinctly presented the rationale for the study in the introduction. The methods are clearly described but would benefit from the inclusion of the CONSORT checklist, in the supplementary section. It would be useful to include information about trial registration. The authors are to be commended in the presentation of the results which are presented primarily as mean differences with 95%CI, eliminating the need to include p-values. The intervention had no effect on the outcomes, even after sub-group analysis was undertaken. The authors complete the paper with a concise but well thought-out discussion. The inclusion of supplemental tables and figures are useful for those who are interested in more detail. A minor point is the use of tense; sometimes present and sometimes past. In conclusion, this paper provides an exemplary model for how clinical researchers can present data from a RCT.

Reviewer #3: The manuscript under review by Strand et al. reports on a randomized controlled trial of B12 supplementation to Nepalese infants and potential effects on metabolic and developmental outcomes. 

The manuscript is nicely written and generally fits the scope of PLoS Medicine, however, the results as presented may not be sufficient to be considered a substantial advance over existing knowledge. While they are clear and support the conclusion of the authors, the results may not be applicable to the general population as the study focus concentrated on marginally stunned children with a specific treatment dose of B12. While there is no question about the validity of this approach, it may have not resulted in improvements for neurodevelopment and growth but the dose was sufficient to improve selective B12 biomarkers in the deficient infant subgroup indicating that there could be a benefit of B12 supplementation. The authors rightfully discussed this topic in the manuscript, thus I would assume that they also consider the possibility that B12 supplementation can be beneficial for developmental markers, possible after dose adjustments. Thus, the current results may not be as exciting for the Bayley-III outcomes, but they are indicative of a selective metabolic response based on B12 status at baseline, which are not unexpected to be noticeable before notable changes in growth and development, and therefore important results for future studies in the field. I encourage to investigate the Bayley-III results between the B12-sufficient infants at baseline and the B12 deficient children to examine possible significant differences in the development tests (and not just exclude B12 sufficient children from the analysis), other than stunting. If the Bayley-III results do not differ by B12 status, then a non-response to the intervention is not surprising, since there wasn't a difference to begin with. 

Moreover, the study included infants with sufficient B12 status, over 100 per group, which changes the sample size, and possibly effect size. We don't know if this may have affected the outcome, but it could be by reducing the sample numbers for certain statistical analyses the effect size just changed enough to not be significant. However, given the limited amount of studies on this topic, this manuscript does add to the knowledge base, but conclusions should be revisited. Additional comments and concerns should be addressed before this manuscript may be recommended for publication:

Line numbers are missing. Please add as per journal guidelines, which is also beneficial for the reviewer. 

Please add a space before reference directly attached to the word, e.g. but not limited to P9 "charts[29]".

Abstract:

RDA is an abbreviation and should be explained at the first use.

As mentioned above, I do not recommend to generalize the results obtained to a population basis as some your results differ depending on the B12 status of the children. 

Author Summary:

The meaning of the findings should be revised based on my discussion above.

Introduction:

P4 last paragraph: A daily oral supplementation with 1.8µg?

P4 last paragraph: I am not quite clear how the results from the 3 RCTs support that the public health concern is not sufficient to alter feeding recommendations, given that all trials showed benefits of the B12 interventions. A little bit more about your thought process here would be beneficial to the reader. 

Subjects and Methods:

P6 Intervention and co-intervention: each sachet provided the daily dose of supplements.

P7 1st line: home visits were conducted by whom?

P8 1st paragraph: please add model and manufacturer with location for the used scales and any equipment used for height measurements. 

P8 first paragraph: please add reference(s) for the mentioned standard guidelines.

P8 1st paragraph: mean values of how many replicate measurements?

P8 Laboratory procedures: HemoCue model?

P8 Laboratory procedures: Temperature during centrifugation? Which model, manufacturer of the centrifuge?

I assume samples were shipped on dry ice to Norway for analysis?

P8 Laboratory procedures: The B12 analysis is not based on the strain of bacteria but the strains were used for this.

P8 Laboratory procedures: "The functional markers plasma homocysteine…". This sentence does not belong in the Methods section, but could be added in the Introduction and/or Discussion.

P9 statistics: Generally, looks good to me but I am not an expert in statistics.

P9 statistics: You defined z-scores -2 based on the WHO growth charts? Please specify or add references. 

P10: Results, 1st paragraph: 572 completed the Bayley-III, but did you also get blood from 572 children as well?

Results:

P11, 2nd paragraph: "We did not find adverse effects…"

Discussion:

P11, Discussion: the 2nd sentence is unnecessary if you have to explain it in the third sentence. Please omit. 

Please add a Discussion about my points in the opening paragraph regarding your results and their interpretation. 

Tables and Figures:

Where applicable, please add to all Tables footnotes and Figure legends the statistical test used to examine significant differences. 

Supplemental Table 3: 1st row = n (children), 2nd row = n (events). OK, but then the next 3 rows, which fall under n(events) appear to be n(children) undergoing the different numbers of events. The specific diagnosis, however, belongs with n(events). This needs to be clarified to the reader.

[LINK]

---

## [Decision Letter · Decision Letter 2]

1 Oct 2020

Dear Dr. Strand,

Thank you very much for re-submitting your manuscript "The effect of vitamin B12 supplementation on neurodevelopment and growth in Nepalese Infants: A randomized controlled trial" (PMEDICINE-D-20-00074R2) for review by PLOS Medicine.

I have discussed the paper with my colleagues and the academic editor and it was also seen again by two reviewers. I am pleased to say that provided the remaining editorial and production issues are dealt with we are planning to accept the paper for publication in the journal.

[LINK]

We look forward to receiving the revised manuscript by Oct 08 2020 11:59PM. 

Sincerely,

Artur Arikainen

Associate Editor 

PLOS Medicine

plosmedicine.org

Requests from Editors:

1. Title: Please update to: “Effects of vitamin B12 supplementation on neurodevelopment and growth in Nepalese Infants: A randomized controlled trial”

2. Financial Disclosure: Please list the funders using full sentences, eg. “This study was funded by…”

3. Data Availability Statement: If the data are not freely available, please describe briefly the ethical, legal, or contractual restriction that prevents you from sharing it. Please also include an appropriate contact (web or email address) for inquiries (this cannot be the study authors).

4. Competing Interests: Please mention the paid or unpaid employment by one or more authors at “Bevital AS, Bergen, Norway”.

5. Abstract: 

a. Around lines 9-11, please give the study recruitment dates.

b. Line 13: Please rephrase as: “The primary outcomes were ...; secondary outcomes included ...".

c. Line 15: Please give the numbers lost to follow-up by assigned group. Delete “Only”.

d. Line 16: Please correct to "94%" to match the main Results. Also, clarify that: "…reported to have been consumed..."

e. Line 17: State the primary outcome results first, even if negative. 

f. Line 18: Delete “substantially”.

g. Please give point effect estimates, 95% CI and p values for all results mentioned, even if not significantly different, including: “The two indirect functional biomarkers of vitamin B12, plasma total homocysteine and methylmalonic acid, were significantly and substantially reduced in the vitamin B12 group compared to the placebo group. There were no effects on the Bayley-III scores, growth, or hemoglobin.”

h. Please mention whether any adverse events were recorded, and whether they were linked to treatment.

i. Line 22: Please mention another limitation.

j. Line 24: Please start with: “In this study, we observed that…”. Please also state the primary outcome first; followed by the other findings in the style "…appeared to result in an improved…”

k. Line 26: Please add a brief sentence on the interpretation or overall relevance of these results.

6. Author Summary:

a. Please use bullet points to break up the paragraphs.

b. Lines 41 and 46: Please perhaps clarify the link between anaemia and hemoglobin concentration for lay readers.

c. Line 46: Please define “subclinical” or remove the term, for clarity to a lay reader.

d. Line 48: Please define “marginally” in his context or remove the term, for clarity to a lay reader.

e. Lines 49-50 and 54: Please define “metabolic profile”, for clarity to a lay reader.

7. Please remove spaces from within citation callouts, eg “…and fish [3,4].”

8. Please replace the word “subjects” with “patients” throughout.

9. Line 114: Please give full recruitment dates, including day.

10. Line 245: Please state the primary outcomes first.

11. Results and Tables: Please include p values alongside 95% CIs, including where not significant.

12. Fig 2: Please describe in the legend what the shaded areas represent.

13. Please ensure all abbreviations are defined in Table footnotes.

14. PLOS does not permit "data not shown.” Please remove this claim, or do one of the following:

a) If you are the owner of the data relevant to this claim, please provide the data in accordance with the PLOS data policy, and update your Data Availability Statement as needed.

b) If the data not shown refer to a study from another group that has not been published, please cite personal communication in your manuscript text (it should not be included in the reference section). Please provide the name of the individual, the affiliation, and date of communication. The individual must provide PLOS Medicine written permission to be named for this purpose.

c) For any other circumstance, please contact me ASAP.

Please rename your CONSORT checklist file “S1 Checklist”.

15. When completing the CONSORT checklist, please use section and paragraph numbers, rather than page numbers.

16. Please mention in your Methods section: “This study is reported according to the CONSORT guidelines (S1 Checklist).”

17. Please rename your analysis plan and protocol documents S1 Text, S2 Text etc. Please mention in your Methods section: “This analyses in this study were carried out according to the predefined protocols and analysis plans (S1-S[n] Texts).” Please also describe any changes to the protocol that took place after the start of the study, and why they were made.

18. Lines 352-375: Please remove the Author Contribution, Conflict of Interest, Funding statements – these should be completed on the online submission form.

19. Line 376: Please rename “Additional Contributions” to “Acknowledgements”.

20. Please provide a URL or DOI for reference 21.

---

Comments from Reviewers:

Reviewer #1: I thank the authors for replying to my comments. I would only make a minor suggestion: instead of writing "crude and multiple generalized linear models" I would suggest "univariate and multivariate generalized linear models". 

Reviewer #3: The authors have carefully addressed my comments, concerns, and questions. One last minor point would be that the calculation and use of 3cB12 was nicely explained in the Methods, there is no actual mentioning of "3cB12" in the Results and Discussion. Figure 2 is the main illustration where cB12 was used, which is the main Figure for End Point results (page 11). For the less familiar reader it would be beneficial to add the "3cB12' to the text to make the conception to the Methods section. Same applies to the Discussion section. Otherwise the manuscript can be considered for publication from my side.

[LINK]

---

## [Editor Report · Decision Letter 3]

23 Oct 2020

Dear Prof. Strand, 

On behalf of my colleagues and the academic editor, Dr. Lars Åke Persson, I am delighted to inform you that your manuscript entitled "Effects of vitamin B12 supplementation on neurodevelopment and growth in Nepalese Infants: A randomized controlled trial" (PMEDICINE-D-20-00074R3) has been accepted for publication in PLOS Medicine. 

PRODUCTION PROCESS

Before publication you will see the copyedited word document (within 5 business days) and a PDF proof shortly after that. The copyeditor will be in touch shortly before sending you the copyedited Word document. We will make some revisions at copyediting stage to conform to our general style, and for clarification. When you receive this version you should check and revise it very carefully, including figures, tables, references, and supporting information, because corrections at the next stage (proofs) will be strictly limited to (1) errors in author names or affiliations, (2) errors of scientific fact that would cause misunderstandings to readers, and (3) printer's (introduced) errors. Please return the copyedited file within 2 business days in order to ensure timely delivery of the PDF proof. 

If you are likely to be away when either this document or the proof is sent, please ensure we have contact information of a second person, as we will need you to respond quickly at each point. Given the disruptions resulting from the ongoing COVID-19 pandemic, there may be delays in the production process. We apologise in advance for any inconvenience caused and will do our best to minimize impact as far as possible.

PRESS

PROFILE INFORMATION

Thank you again for submitting the manuscript to PLOS Medicine. We look forward to publishing it. 

Best wishes, 

Emma Veitch, 

Senior Editor 

PLOS Medicine

plosmedicine.org